# Results of the phase IIa RADICAL trial of the FGFR inhibitor AZD4547 in endocrine resistant breast cancer

R. C. Coombes [1✉], P. D. Badman [1], J. P. Lozano-Kuehne [1], X. Liu[1], I. R. Macpherson [2], I. Zubairi[2], R. D. Baird [3], N. Rosenfeld [3], J. Garcia-Corbacho [3], N. Cresti [4], R. Plummer[4], A. Armstrong[5], R. Allerton[6], D. Landers [7], H. Nicholas[1], L. McLellan[8], A. Lim[9], F. Mouliere [3,10], O. E. Pardo [1], V. Ferguson[1] & M. J. Seckl [1✉]

We conducted a phase IIa, multi-centre, open label, single arm study (RADICAL; NCT01791985) of AZD4547 (a potent and selective inhibitor of Fibroblast Growth Factor Receptor (FGFR)-1, 2 and 3 receptor tyrosine kinases) administered with anastrozole or letrozole in estrogen receptor positive metastatic breast cancer patients who had become resistant to aromatase inhibitors. After a safety run-in study to assess safety and tolerability, we recruited 52 patients. The primary endpoint was change in tumour size at 12 weeks, and secondary endpoints were to assess response at 6 weeks, 20 weeks and every 8 weeks thereafter and tolerability of the combined treatment. Two partial responses (PR) and 19 stable disease (SD) patients were observed at the 12-week time point. At 28 weeks, according to centrally reviewed Response Evaluation Criteria in Solid Tumours (RECIST) criteria, five PR and 8 SD patients were observed in 50 assessable cases. Overall, objective response rate (5 PR) was of 10%, meeting the pre-specified endpoint. Fourteen patients discontinued due to adverse events. Eleven patients had retinal pigment epithelial detachments which was asymptomatic and reversible in all but one patient. Exploratory ribonucleic acid sequencing (RNA-Seq) analysis was done on patients' samples: 6 differentially-expressed-genes could distinguish those who benefited from the addition of AZD4547.

[1] Department of Surgery and Cancer, Imperial College London, London, UK. [2] Cancer Research UK Clinical Trials Unit, Beatson West of Scotland Cancer Centre, Glasgow, UK. [3] Medical Oncology, Addenbrooke's Hospital, Breast Cancer Research Unit, Cancer Research UK Cambridge Centre, Cambridge, UK. [4] Sir Bobby Robson Cancer Trials Research Centre, Northern Centre for Cancer Care, Freeman Hospital, Newcastle, UK. [5] Breast Research Office, The Christie NHS Foundation Trust, Christie Hospital, Manchester, UK. [6] C8 Admin Offices, Russell's Hall Hospital, Russells Hall, UK. [7] Astrazeneca, Cheshire, UK. [8] ECMC Programme Office, Research and Innovation, Cancer Research UK, London, UK. [9] Department of Metabolism, Digestion and Reproduction, Imperial College London, London, UK. [10] Amsterdam UMC, Vrije Universiteit Amsterdam, Department of Pathology, Cancer Centre Amsterdam, Amsterdam, The Netherlands. ✉email: c.coombes@imperial.ac.uk; m.seckl@imperial.ac.uk

Breast cancer is one of the most commonly diagnosed cancers worldwide and the second leading cause of cancer-related deaths in women[1]. The treatment of breast cancer is determined by the extent of the disease and a variety of other prognostic factors, including hormone receptor status. The most important factor determining response to hormonal manipulation is the presence of the oestrogen receptor (ER) in the target tissue[2].

At the time of inception of this study, the choice of first-line endocrine therapy for metastatic breast cancer was generally considered to be anti-oestrogen monotherapy such as Faslodex or a non-steroidal aromatase inhibitor (NSAI) such as anastrozole or letrozole. Options for subsequent endocrine therapy included a steroidal aromatase inhibitor (AI) such as exemestane[3]. Irrespective of the treatment sequence, nearly all patients ultimately experience disease progression and therefore there remains a need to identify further treatment options for those patients who progress.

FGFR 1 is overexpressed in breast cancers and its expression is related to prognosis[4]. Further, addition of FGF-2 to breast cancer cell lines in vitro impairs the effects of non-steroidal AIs and tamoxifen whilst downregulation of FGFR1 by siRNAs sensitises breast cancer cells to these agents[5].

AZD4547 is a potent and selective inhibitor of FGFR-1, 2 and 3 receptor tyrosine kinases. Phase 1 studies both in Europe and Japan[6,7] have shown that the compound is well tolerated and active in patients with solid tumours. Here, we hypothesised that AZD4547 could reverse resistance to AIs such as anastrozole and letrozole. Non-published single agent data with AZD4547 in ER positive breast cancer had already shown little efficacy and for this reason we chose to combine the AI with AZD4547.

In addition, we hoped to determine whether amplification of FGFR1 or expression of certain genes could distinguish those patients who could benefit from AZD4547.

Here we show that, in patients with ER positive breast cancer who have progressed on aromatase inhibitors, the addition of the FGFR inhibitor gives clinical benefit, irrespective of FGFR amplification, but especially in those whose breast cancers display a pattern of gene expression linked to FGF action.

## Results

**Safety run-in.** Between October 2012 and August 2013, a total of 6 patients were recruited into the safety run-in part of the trial. Three received anastrozole 1 mg and the remainder letrozole 2.5 mg daily continuously all combined with AZD4547 80 mg on a one week on one week off schedule. All were heavily pre-treated ER positive breast cancer patients. Compliance with taking AZD4547 was generally good, monitored by patient diary and counting remaining tablets. Two patients stopped for 9 and 13 days, due to colonic obstruction and retinal pigment epithelial detachment (RPED), respectively. These AEs spontaneously resolved and normal dosing was resumed. No DLTs were observed but all 6 patients reported at least one grade 1/2 AE including rash, raised liver function tests and with 5 having increased blood phosphate levels, an expected on-target effect of FGFR inhibition. Serial serum FGF2 and FGF23 levels demonstrated a fall in FGF2 at Cycle 1 day 7 but no discernible change in FGF23 on AZD4547 therapy. Importantly, PK measurements (see Supplementary Note 1) showed no evidence of interactions between AZD4547 and either of the NSAI's. Therefore, the SRC determined that this dose and schedule should be taken forward into Phase IIa.

**Phase IIa study population.** Between April 2014 and December 2015, we recruited 52 postmenopausal women with ER+ breast

cancer who had progressed on treatment with either anastrozole or letrozole in any metastatic setting. The Consort diagram shows the patient disposition (Supplementary Fig. 1) and explains why two more patients than originally intended were recruited.

Table 1 shows the baseline characteristics for the study population. Forty-nine of the 52 enroled patients had previously

### Table 1 Patient baseline characteristics[a].

| | Letrozole N = 49 | Total N = 52 |
|---|---|---|
| Age in years, median (IQR) | 56(50–64) | 56.5(50–64) |
| Ethnicity | | |
| White | 46 (93.9%) | 49 (94.2%) |
| Black | 2 (4.1%) | 2 (3.8%) |
| Not reported | 1 (2.0%) | 1 (1.9%) |
| ECOG status | | |
| 0 – Fully active | 31 (63.3%) | 33 (63.5%) |
| 1 – Restricted in physically strenuous activity | 18 (36.7%) | 19 (36.5%) |
| Endocrine[b] | | |
| Anastrozole | 2 (4.1%) | 5 (9.6%) |
| Letrozole | 49 (100%) | 50 (96.2% |
| Tamoxifen | 35 (71.4%) | 38 (73.1%) |
| Everolimus | 3 (6.1%) | 3 (5.8%) |
| Exemestane | 13 (26.5%) | 14 (26.9%) |
| Other | 6 (12.2%) | 6 (11.5%) |
| Prior radiotherapy (yes) | 35 (71.4%) | 37 (71.2%) |
| Primary tumour only | 19 | 20 |
| With metastasis | 16 | 17 |
| Prior targeted therapy (yes) | 7 (14.3%) | 7 (13.5%) |
| Primary tumour only | 2 | 2 |
| With metastasis | 5 | 5 |
| Prior chemotherapy (yes) | 41 (83.7%) | 43 (82.7%) |
| Primary tumour only | 23 | 23 |
| With metastasis | 18 | 20 |
| Prior surgery for cancer (yes) | 40 (81.6%) | 42 (80.8%) |
| Primary tumour only | 21 | 21 |
| With metastasis | 19 | 21 |
| Nos. of systemic therapies after progression on NSAI before study | | |
| 0 | 27 (55.1%) | 28 (53.8%) |
| 1 | 5 (10.2%) | 6 (11.5%) |
| 2 | 7 (14.3%) | 7 (13.5%) |
| 3 | 2 (4.1%) | 2 (3.8%) |
| 4 | 6 (12.2%) | 6 (11.5%) |
| 6 | 2 (4.1%) | 2 (3.8%) |
| 7 | 0 (0%) | 1 (1.9%) |
| Tumour grade[c] | | |
| G1/2 | 24 (49.0%) | 25 (48.1%) |
| G3 | 13 (26.5%) | 14 (26.9%) |
| Unknown | 12 (24.5%) | 13 (25.0%) |
| Stage (Primary) | | |
| Stage 1–3 | 25 (51.0%) | 25 (48.1%) |
| Stage 4 | 24 (49.0%) | 27 (51.9%) |
| ER status[c] | | |
| Positive | 49 (100%) | 52 (100%) |
| PgR status[c] | | |
| Positive | 20 (40.8%) | 20 (38.5%) |
| Negative | 12 (24.5%) | 12 (23.1%) |
| Unknown | 17 (34.7%) | 20 (38.5%) |
| HER2 status (by IHC or FISH)[c] | | |
| Positive | 2 (4.1%) | 3 (5.8%) |
| Negative | 41 (83.7%) | 43 (82.7%) |
| Unknown | 6 (12.2%) | 6 (11.5%) |

[a]Data are presented as frequency (percentage) for categorical variables and median (IQR) for continuous variables.
[b]Patients may have had more than 1 endocrine treatment in neoadjuvant or adjuvant setting.
[c]Tumour characteristics for primary or metastatic tumour; For HER2 status by either IHC or FISH, the FISH result supersedes the IHC.

**Table 2 Most frequently observed treatment-emergent adverse events[a].**

| Treatment-emergent adverse event (AE) | Severity | | | | | | | | Total | | | |
| --- | --- | --- | --- | --- | --- | --- | --- | --- | --- | --- | --- | --- |
| | Grade 1–2 | | | | Grade 3 | | | | Events (N = 505) | | Patients (N = 52) | |
| | Events | | Patients | | Events | | Patients | | | | | |
| Hyperphosphataemia | 43 | 8.5% | 26 | 50% | 0 | | 0 | | 43 | 8.5% | 26 | 50% |
| Dry mouth | 29 | 5.7% | 23 | 44% | 0 | | 0 | | 29 | 5.7% | 23 | 44% |
| Alopecia | 23 | 4.6% | 20 | 38% | 0 | | 0 | | 23 | 4.6% | 20 | 38% |
| Dysgeusia | 15 | 3.0% | 13 | 25% | 0 | | 0 | | 15 | 3.0% | 13 | 25% |
| Constipation | 12 | 2.4% | 12 | 23% | 0 | | 0 | | 12 | 2.4% | 12 | 23% |
| Nausea | 14 | 2.8% | 12 | 23% | 0 | | 0 | | 14 | 2.8% | 12 | 23% |
| Retinal pigment epithelium detachment (RPED) | 16 | 3.2% | 11 | 21% | 0 | | 0 | | 16 | 3.2% | 11 | 21% |
| Diarrhoea | 16 | 3.2% | 10 | 19% | 0 | | 0 | | 16 | 3.2% | 10 | 19% |
| Dyspepsia | 14 | 2.8% | 10 | 19% | 0 | | 0 | | 14 | 2.8% | 10 | 19% |
| Decreased appetite | 14 | 2.8% | 8 | 15% | 0 | | 0 | | 14 | 2.8% | 8 | 15% |
| Dry eye | 9 | 1.8% | 8 | 15% | 0 | | 0 | | 9 | 1.8% | 8 | 15% |
| Dry skin | 9 | 1.8% | 8 | 15% | 0 | | 0 | | 9 | 1.8% | 8 | 15% |
| Epistaxis | 11 | 2.2% | 8 | 15% | 0 | | 0 | | 11 | 2.2% | 8 | 15% |
| Stomatitis | 10 | 2.0% | 8 | 15% | 0 | | 0 | | 10 | 2.0% | 8 | 15% |
| Alanine aminotransferase increased | 11 | 2.2% | 7 | 13% | 0 | | 0 | | 11 | 2.2% | 7 | 13% |
| Aspartate aminotransferase increased | 9 | 1.8% | 6 | 12% | 1 | 0.2% | 1 | 2% | 10 | 2.0% | 7 | 13% |
| Blood calcium increased | 9 | 1.8% | 7 | 13% | 0 | | 0 | | 9 | 1.8% | 7 | 13% |
| Calcium phosphate product increased | 11 | 2.2% | 7 | 13% | 0 | | 0 | | 11 | 2.2% | 7 | 13% |
| Fatigue | 12 | 2.4% | 7 | 13% | 1 | 0.2% | 1 | 2% | 13 | 2.6% | 7 | 13% |
| Nail disorder | 7 | 1.4% | 7 | 13% | 0 | | 0 | | 7 | 1.4% | 7 | 13% |
| Onycholysis | 6 | 1.2% | 6 | 12% | 0 | | 0 | | 6 | 1.2% | 6 | 12% |
| Arthralgia | 5 | 1.0% | 5 | 10% | 0 | | 0 | | 5 | 1.0% | 5 | 10% |
| Blood albumin decreased | 12 | 2.4% | 5 | 10% | 0 | | 0 | | 12 | 2.4% | 5 | 10% |
| Blood alkaline phosphatase increased | 5 | 1.0% | 4 | 8% | 1 | 0.2% | 1 | 2% | 6 | 1.2% | 5 | 10% |
| Glossodynia | 5 | 1.0% | 5 | 10% | 0 | | 0 | | 5 | 1.0% | 5 | 10% |
| Lethargy | 5 | 1.0% | 5 | 10% | 1 | 0.2% | 1 | 2% | 6 | 1.2% | 5 | 10% |
| Mouth ulceration | 13 | 2.6% | 5 | 10% | 0 | | 0 | | 13 | 2.6% | 5 | 10% |
| Vomiting | 6 | 1.2% | 5 | 10% | 0 | | 0 | | 6 | 1.2% | 5 | 10% |
| Any treatment-emergent AE[b] | 496 | 98.2% | 50 | 96% | 7 | 1.4% | 6 | 12% | 505 | 100% | 50 | 96% |

[a]Treatment-emergent adverse events (AEs) occurring at any time point, in at least 10% of patients and are categorised as "definitely", "possibly" or "probably" related to the study treatment. Out of the 822 AEs experienced by 52 study participants, 505 are classified as treatment-emergent AEs. There are study participants with repeated or multiple adverse events. Source data are provided as a Source Data file.
[b]There are 2 adverse events out of the total 505 treatment-emergent adverse events without severity grading. The severity of treatment-emergent AEs observed are only from grades 1–3. Of the 52 patients who reported experiencing AEs, 50 of them had treatment-emergent AE

progressed on Letrozole and 3 on Anastrozole with 82% having also received prior chemotherapy. Just over 50% of the patients proceeded straight from AI failure into the trial the remainder having between 1–6 other systemic therapies before study entry. Original tumour stage and grade were mostly 1–3 and 1–2 but was missing in 12 and 13% of patients respectively. None of the patients had clinically significant baseline ECG or ocular abnormalities.

**Adverse events and study compliance**. All 52 patients experienced AEs (total 822 recorded AEs) most of which were CTCAE ≤ 2. Common adverse events (grade ≤ 2) included raised serum phosphate (26), dry mouth (30), diarrhoea (30), nausea (26), constipation (23) and fatigue (20) Fourteen patients discontinued due to significant AEs. There were 36 severe adverse events (CTCAE grade ≥ 3) including corneal ulcer, oesophageal mucositis, elevated liver enzymes (AST/ALP), anaemia and fatigue/lethargy. All the foregoing only affected a single patient with the exception of thrombocytopenia ($n = 3$; 8.3%), increased AST

($n = 2$; 5.6%), lymphopenia ($n = 2$; 5.6%) and neutropenic sepsis ($n = 2$; 5.6%).

Treatment-emergent effects are shown in Table 2. Elevation of phosphate levels were seen in 26 patients (Fig. 1) and elevated calcium in 7. Treatment delays were common due to the need to normalise blood chemistry. Eleven patients had RPEDs either in one or both eyes. Table 3 shows the time to onset of RPED after starting AZD4547, its duration and outcome. Ten patients demonstrated improved or completely resolved RPED during the study follow-up and there was no recurrent RPED. In one case, RPED improved after cessation of treatment but recurred when therapy was re-introduced. The treatment was then permanently discontinued. The median time to diagnosis of RPED in the study was 1 month (Range: 0.8–6.7) and the median duration was 0.7 month (0.4–2.1). Importantly, no patients experienced symptoms related to their RPED. Nevertheless, six patients with asymptomatic RPED at a single enroling centre discontinued treatment on the advice of their attending oncologist.

Drug compliance data for both the AI and AZD4547 were available for 47 patients. The median and IQR of compliance was

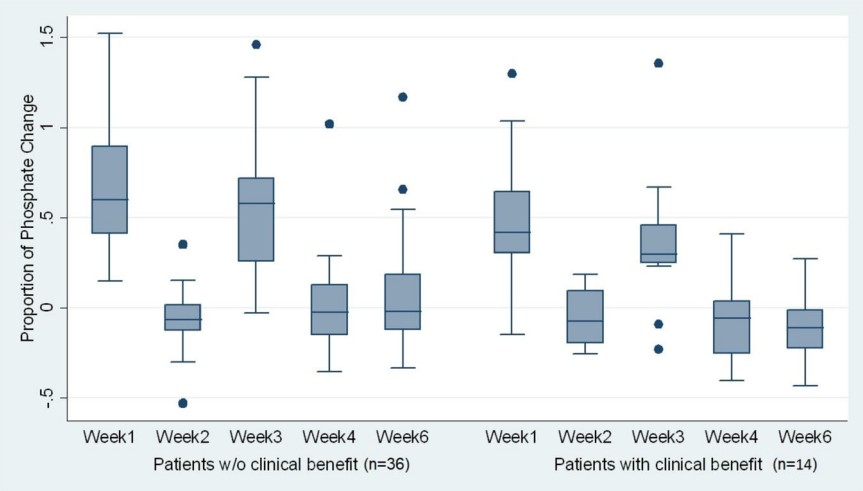

**Fig. 1 Boxplots of phosphate changes, showing the proportion of phosphate change at different time points using cycle 1 day 1 phosphate as baseline.** Two patients without follow-up data are not included in the graph. The centre lines are defined by the median. The 'outsider' dots are the data points that fall outside the line called "whiskers" in the boxplot. The whiskers include all data points within 1.5 IQR (Interquartile range) of the nearer quartile (e.g. 3rd Quartile or 75th percentile). Source data are provided as a Source Data file.

100% (99.1–100%) and 99.4% (96.2–100%) for AI and AZD4547, respectively. By July 2018, all 52 enroled patients had discontinued study medication. Thirty-five discontinued due to disease progression, 14 due to AEs, 2 due to patient decision and 1 due to investigator's decision. Supplementary Table 2 shows the details of the patients who permanently discontinued the study treatment, including the reason for discontinuation and their tumour response.

**Clinical efficacy**. Central RECIST review was carried out in 50 of 52 patients. Table 4 and Fig. 2 show the summary of tumour size changes and response at weeks 6, 12, 20 and 28 and the Waterfall plot, respectively, based on central imaging review. A 'Swimmer Plot' is also shown (Fig. 2a, b).

The primary endpoint (response at week 12) showed two partial responses (PR) and 19 stable disease (SD) patients. The Clinical Benefit Rate (CBR = PR or SD, as defined in the Interim Analysis section of the Study Protocol) at week 12 was 40.4% (95% CI: 27.0–54.9%) by central review.

At 28 weeks, three additional patients' cancers achieved a PR (6%) in addition to the earlier 2 PRs and 8 showed stable disease (SD). There were no complete responses; 14 other patients showed reduction in size of measurable metastases (see Table 4 and Swimmer and Waterfall plots, Fig. 2).

Overall, five partial responses (PR) were confirmed by central review (10%); thus, the objective response rate was 5 of 50 patients (10%) and the CBR after 28 weeks on treatment was 13 (26%; 95% CI: 14.6–40.3%) of 50 assessable patients and on an intention-to-treat basis was 13 of 52 or 25% (95% CI: 14.0–38.9%).

The median progression-free-survival for all patients was 92 days (95% CI: 71–185), for patients in the CBR group was 137 days (range: 32–1266 days).

Twenty-eight (53.8%) out of 52 enroled patients entered the study after failing their aromatase inhibitor (AI) Letrozole or Anastrozole, with no intervening therapy, and 7 (25%) experienced clinical benefit (PR or SD). The remaining patients had a median of 2 (range 1–7) of other lines of therapy before retreatment with their failed AI plus AZD4547 and 6 (25%) benefited, thus the response rates were no different between these two groups of patients.

**Exploratory analyses - biomarkers of response**. We examined whether increased serum phosphate (obtained from all patients in the phase 2 study) as a marker of AZD4547 target engagement might correlate with response. Increased phosphate levels were observed when patients received AZD4547 and a fall in the week off therapy, but there was no relationship between extent of phosphate change and response to treatment (Fig. 1). We also examined whether the development of RPED correlated with response, but again found no evidence for this. Thus, RPED developed in 3 of 13 patients (23.1%) with clinical benefit (PR and SD) and in 5 of 32 (15.6%) without clinical benefit (p = 0.55). The other three patients who had RPED had unassessable clinical benefit (Table 3).

In addition, we measured FGFR copy number in the plasma of study patients. Only 4 showed FGFR1 amplification and there was no relationship between this and response. However, we used t-MAD which is a summary metric, and has been shown to be capable of estimating the ctDNA tumour fraction in response to treatment[8]. The overall tumour fraction in the plasma samples was determined by shallow Whole Genome Sequencing (sWGS) and calculated using the t-MAD score (see Methods). The week 6 and week 8 on-treatment ctDNA t-MAD scores were associated with subsequent RECIST response on imaging (Fig. 3). A significant enrichment in the tumour fraction observed with t-MAD was observed in patient with PD in comparison to SD (Wilcoxon, p = 0.0017) and with patients with PR (Wilcoxon, p = 0.0018) for the cases collected ~4 weeks after treatment. We also observe a non-significant increase in tumour fraction between SD and PR patients ~4 weeks after treatment (Wilcoxon, p = 0.088). At ~6 weeks the ctDNA fraction was significantly decreased in responding patients (Wilcoxon, p = 0.017 when compared to SD and p = 0.01 when compared to PD).

We also measured the expression of 2549 genes involved in cancer progression in tumour sections obtained from 16 primary and 4 metastatic samples taken at diagnosis from 12 patients with stable (SD) and 6 with progressive disease (PD) using the HTG EdgeSeq Oncology Biomarker Panel (Supplementary Fig. 2a–d). This enabled us to identify 35 genes as differentially expressed in SD vs PD patients (p < 0.05, FDR < 0.1) (Supplementary Fig. 3a). However, hierarchical clustering based on the expression data for these differentially expressed genes (DEGs) failed to satisfactorily

**Table 3 Description of RPEDs (retinal pigment epithelial detachments).**

| Subject number | Right eye | | | Left eye | | | Reason for permanent treatment discontinuation |
|---|---|---|---|---|---|---|---|
| | Time to diagnosis (month) | Duration (month) | Outcome | Time to diagnosis (month) | Duration (month) | Outcome | |
| RAD01-20002 | 6.7 | 0.5 | Improved | 6.7 | 0.5 | Improved | AE – not RPED |
| RAD01-20005 | 1.0 | 0.5 | Completely resolved | 1.0 | 0.5 | Completely resolved | Disease Progression |
| RAD01-20013 | 1.9 | 1.2 | Worsened when drug was re-introduced | 1.9 | 1.2 | Worsened when drug was re-introduced | AE – RPED |
| RAD03-20003 | | | | 1.9 | 0.5 | Completely resolved | Disease Progression (*With temporarily treatment discontinuation due to AE including RPED) |
| RAD03-20005 | 0.9 | 0.5 | Completely resolved | | | | Disease Progression |
| RAD04-20004 | 2.0 | 0.8 | Completely resolved | 0.9 | 1.9 | Completely resolved | AE – including RPED |
| RAD04-20005 | 0.8 | 0.9 | Improved | 0.8 | 0.9 | Improved | AE – including RPED |
| RAD04-20009 | 0.8 | 0.4 | Improved | | | | AE – including RPED |
| RAD04-20010 | 0.9 | 2.1 | Completely resolved | 0.9 | 2.1 | Completely resolved | AE – including RPED |
| RAD05-20005 | 1.9 | 1.2 | Completely resolved | | | | Disease Progression |
| RAD05-20006 | | | | 2.3 | 0.7 | Completely resolved | AE – not RPED |

separate PD from SD patients (Supplementary Fig. 3). We therefore performed principal component analysis (PCA) that segregated PR + PD verses SD patients based on the expression levels of our DEGs (Supplementary Fig. 2a, b) as compared to PCA performed using the entire dataset (Supplementary Fig. 3c). However, the spread of the data was still suboptimal, with one SD patient clearly clustering with PD patients. Segregation was further improved by performing the PCA only on the top 6 overexpressed DEGs, with the PD patients now very tightly clustered, providing superior separation between PD verses PR + SD patients (Supplementary Fig. 2a-right). This improvement was not the result of overfitting due to the low number of genes included in the analysis as no such improvement, or indeed separation between PR + SD verses PD patients, was observed when performing the PCA on the 6 top under-expressed DEGs (Supplementary Fig. 3d). Hence, it appears that a gene signature composed of the 6 top overexpressed genes in PR + SD samples, namely CHGA, FGF10, PTPRC, MIA, TRIM72 and SEC14L2 (Supplementary Fig. 2b), may predict benefit of patients to our combination therapy. Functional network building followed by gene ontology and pathway enrichment analysis (Supplementary Fig. 2c–e) suggested that several of our DEGs, including 2 of our 6 top overexpressed hits, are principally involved in FGFR signalling which may explain their association with response status to our combination therapy. In addition, our DEGs also associated with positive regulation of ERK signalling and negative regulation of WTN signalling, apoptosis, mitotic spindles and hypoxia. Comparison of the expression of our DEGs between our dataset and the publicly-available GDC TCGA breast cancer and METABRIC datasets revealed that expression correlations between DEGs was partially conserved between all datasets, suggesting that our expression profile is not majorly biased (Supplementary Fig. 4a–c). Although the expression of 5 of our 6-top overexpressed DEGs showed significant positive correlation in our dataset (Supplementary Fig. 4d), this is likely the result of overfitting due to our small sample size. Indeed, such correlation

was not found in the TCGA dataset (Supplementary Fig. 4e), so that expression of each gene in our signature probably acts independently to promote patients' response, a hypothesis supported by their distribution across different biological processes in our functional network (Supplementary Fig. 2c-asterix). Interestingly, analysis of TCGA data revealed that expression of our signature genes was often strongly correlated with only one of either the aromatase genes (CYP19A1, aka ARO), FGFR1 or FGFR2 (Supplementary Fig. 2f and 5a), with the exception of CHGA which correlated with both ARO and FGFR1. In addition, expression of 3 of our 6-top overexpressed DEGs positively correlated with that of SPRY2 in the FGFR pathway (Supplementary Fig. 5b). Taken together, our results suggest that this 6 gene expression signature might aid selection of patients for combined AI plus AZD4547 but this requires further validation.

**Discussion**

This study demonstrates that adding AZD4547 to AI treatment in patients who have become resistant to this treatment can be beneficial in a subset of patients. At week 28, there were 3 patients whose cancers had partially responded in addition to the 2 earlier responders and 8 whose disease had stabilised, giving a response rate of 10% and a CBR of 26%. This finding implies that FGF signalling can cause resistance to AI therapy.

The adverse events observed in this study were mostly (95%) grades 1 and 2 AEs. The AEs with grades 3 or higher accounted for 5% of all AEs. The study observed 13 SAEs, but none were related to the study medication. The adverse events were similar to those noted in the study of Saka et al.[6] and we also observed retinal abnormalities which resulted in discontinuation of treatment in 6 patients. However, in retrospect, discontinuation may not have been needed as all patients were asymptomatic and in those patients where treatment was restarted RPED was a recurrent problem in a single patient. This result is similar to the reported results of GLOW (https://clinicaltrials.gov/ct2/show/results/NCT01202591). Other than discontinuation of therapy,

**Table 4 Summary of tumour assessment based on central review[a].**

|  | Baseline | 6 weeks | 12 weeks | 20 weeks | 28 weeks[b] |
|---|---|---|---|---|---|
| Sum of the diameters for all target lesions (mm)[c] | 67 (28–86) [n = 47] | 73 (31–91) [n = 45] | 61 (24–96) [n = 27] | 53 (20–76) [n = 17] | 32 (23–83) [n = 13] |
| Proportion of Tumour size change[d] |  | 0.10 (0.31) (95% CI: 0.003–0.19) Range: −0.32–1.68 [n = 45] | 0.18 (0.44) (95% CI: 0.03–0.33) Range: −0.40–1.79 [n = 36] | 0.21 (0.45) (95% CI: 0.05–0.37) Range: −0.32–1.79 [n = 32] | 0.22 (0.49) (95% CI: 0.03–0.40) Range: −0.55–1.79 [n = 30] |
| Overall response (n = 52) |  |  |  |  |  |
| Complete response |  | 0 (0%) | 0 (0%) | 0 (0%) | 0 (0%) |
| Partial response |  | 0 (0%) | 2 (3.8%) | 3 (5.8%) | 3 (5.8%) |
| Stable disease |  | 37 (71.2%) | 19 (36.5%) | 13 (25.0%) | 8 (15.4%) |
| Progressive disease |  | 9 (17.3%) | 6 (11.5%) | 2 (3.8%) | 2 (3.8%) |
| Progression before scan |  | 1 (1.9%) | 16 (30.8%) | 25 (48.1%) | 29 (55.8%) |
| Withdrawn before scan |  | 3 (5.8%) | 6 (11.5%) | 7 (13.5%) | 8 (15.4%) |
| Missing |  | 2 (3.8%) | 3 (5.8%) | 2 (3.8%) | 2 (3.8%) |
| Objective Response Rate, ORR (CR + PR) |  | 0 | 2 (95% CI: 0.47–13.21) | 3 (95% CI: 1.21–15.95) | 3 (95% CI: 1.21–15.95) |

[a]Data are presented as frequency (percentage) for categorical variables mean (SDev) for continuous variables; Two patients have no data for central review. Some have no scans in specific timepoints. Source data are provided as a Source Data file.
[b]The clinical benefit rate (CBR) was defined (in the Interim analysis section of the Study Protocol) as CR and PR at any time during the study follow-up and SD for 28 weeks. The CBR at week 28 in this study is 13 out of 50 assessable subjects or 26% (95% CI: 14.6–40.3%).
[c]Data are presented as median (IQR).
[d]Mean change (SDev) in tumour size at a follow-up time is defined as a proportion change in the sum of the diameters for all target lesions at that follow-up time (or progression if prior to that follow-up time) compared to baseline.

no specific treatment was given for RPED, however the need for detailed ophthalmic investigation clearly complicates AZD4547 FGFR inhibitor therapy. Hyperphosphatemia was also commonly observed, necessitating temporary treatment discontinuation. In all cases, phosphate levels returned to normal on discontinuation. Seven patients became hypercalcaemic, also necessitating treatment discontinuation. Neither events seemed more common in those patients who appeared to derive benefit from therapy with AZD4547.

Approximately 8–20% of breast cancers display FGFR1 amplification which correlates with early relapse and poor survival particularly in ER positive breast cancer[5]. This suggests that patients with ER positive breast cancer that have FGFR1 amplification might benefit from FGFR inhibitor-based therapies. However, in our study FGFR amplification in ctDNA did not correlate with response. This appears to concur with another study Van Cutsem et al.[9] that failed to observe a relationship between response and FGFR2 amplification in gastric cancer. Similarly, a study of AZD4547 in FGFR1 amplified lung cancer[10] failed to see responders in the FGFR1 amplified group. Lucitanib[11] has been evaluated in patients with ER positive breast cancer; here, the response rate was 25% in FGFR-high breast cancers (by immunohistochemistry) compared with 8% in low-FGFR1 tumours, but the numbers of patients were small and, although suggestive of a relationship, this result was not conclusive. A recent study showed a correlation between response and FGFR1 mRNA levels[12], but this finding requires confirmation.

To identify other ways to select ER positive breast cancer patients for FGFR inhibitor therapy, we initially examined changes in ctDNA using the t-MAD score. The t-MAD score is a recently developed method converting somatic copy number aberration as a quantitative estimation of the ctDNA fraction in plasma[8]. Our results suggest that patients with persistent high on-treatment t-MAD scores in ctDNA obtained some weeks after the start of treatment might be advised to stop treatment and change treatment (Fig. 3). Thus, rapid decrease in ctDNA tumour fraction from blood might be an early indicator of benefit from the combination, especially if observed on multiple timepoints as indicated by the significant p values at week 6 and 8 in our plot, in comparison to baseline. However, ctDNA tumour fraction is only one marker of response and should be used in conjunction with other clinical parameters.

In order to determine whether a gene expression signature could predict benefit from AZD4547 therapy, we carried out gene expression analysis of tissue sections obtained from our study patients at their original breast cancer diagnosis. Strikingly, we found a set of DEGs, many of which are involved in FGFR signalling, that distinguished responders (PR plus SD) from non-responders (PD) (Supplementary Fig. 2). The fact that expression of our 6 top-discriminating DEGs do not consistently associate with that of both ARO and FGFRs may explain the failure of FGFRs amplification status alone in predicting response to our combination. However, their association with the expression of individual FGFR pathway members may justify their involvement. Taken together, our data suggest that an expression gene signature for CHGA, FGF10, PTPRC, MIA, TRIM72 and SEC14L2 may predict benefit from combined ARO/FGFR inhibition. The reproducibility of this signature needs to be confirmed in a validation cohort. Hence, it should currently be viewed as a starting point for further biomarker development and future studies should optimally be carried out using biopsies obtained before commencing the FGFR inhibitor. However, the fact that expression correlations between these genes and other DEGs are conserved between the TCGA or METABRIC datasets and our own cohort suggests that the expression of these genes is not particularly biased in our patient population, which should maximise chances of future validation.

FGFR inhibitors have been evaluated in ER positive breast cancer by other groups. The combination of Dovitinib, an inhibitor of FGFR1-3, with Fulvestrant in ER positive metastatic breast cancer has been reported by Musolino et al.[13]. In this study, there was no difference in PFS between Dovitinib and placebo in the entire cohort, but the PFS in the FGF pathway amplified group was twice that of the FGF pathway non-amplified group. A further FGFR inhibitor, Erdafitinib, is now licenced in bladder cancer following a successful trial[14] and, to our knowledge, is the only FGFR inhibitor to be licensed for use in oncology so far, and is being evaluated in a combination with palbociclib and fulvestrant in breast cancer in other trials at the present time.

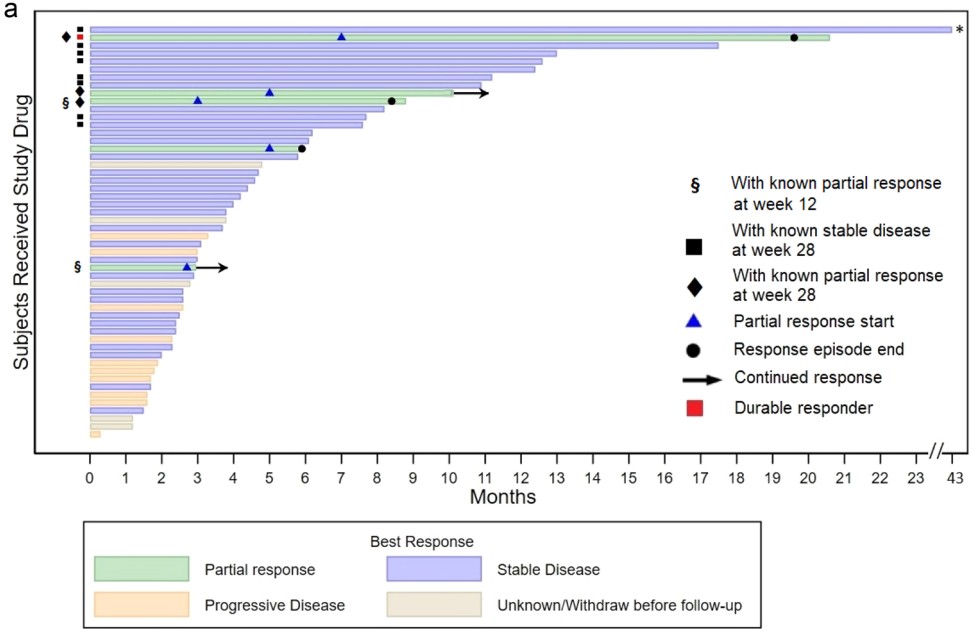

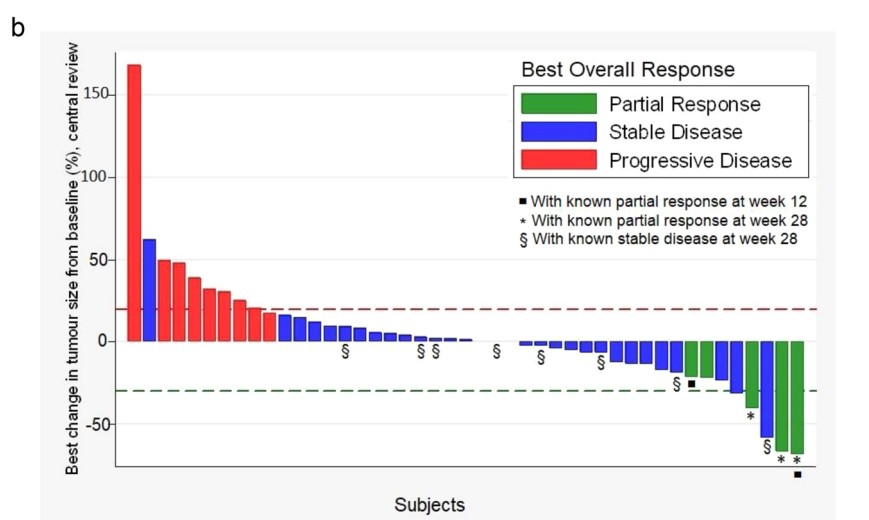

**Fig. 2 AZD4547 in metastatic breast cancer: time on treatment and best overall response. a** Time on Treatment. Each bar represents one subject in the study and their length of treatment in months. *Subject with the longest follow-up has 43 months of treatment duration. § Two subjects had known partial response at week 12 follow-up. Eight subjects (black squares) had known stable disease at week 28 (or 7th month). Three subjects (black diamonds) had known partial response at week 28, while two subjects had known partial response at an earlier follow-up time. During the whole study period, 5 subjects had partial response at any time point. A durable responder (red square) is a subject who has confirmed response for at least 6 months. The two subjects marked with continued response (black arrows) had partial response as their last known response prior to their withdrawal from the study. It is not known when the partial response in these subjects ended. Source data are provided as a Source Data file. **b** Treatment Response – Waterfall Plot. Best overall response and maximum percentage of tumour reduction based on central review of subjects with at least two tumour measurements ($n = 45$); Each bar represents one subject in the study. Subjects with only baseline tumour measurement or who have withdrawn before follow-up are not included in the graph. Source data are provided as a Source Data file. Two subjects (black squares) had partial response at week 12. *Three subjects had partial response at week 28 among the five subjects who experienced partial response at any time point during the study. Two of the subjects with partial response (green bars without *) have stopped treatment prior to week 28. § Eight subjects had stable disease at week 28.

RPED was the adverse event that seemed to differentiate AZD4547 from other inhibitors, in which cardiovascular events such as hypertension seem more common (for a review see ref. [15]).

Recently genetic abnormalities in the FGFR signalling pathway have been found in 40% of patients who have become resistant to endocrine therapy[16]. In addition, because combined therapy with CDK4/6 inhibitors and endocrine therapy are now frequently used as first-line treatment of patients with metastatic disease, an important area for therapeutic intervention is in patients who have become resistant to these treatments. A recent study showed that, as judged by ctDNA analysis, 41% of patients who had become resistant to CDK4/6 inhibitors had FGFR abnormalities[17]. As a result of these observations, trials are in progress to evaluate FGFR inhibitors and are summarised in a recent review[14].

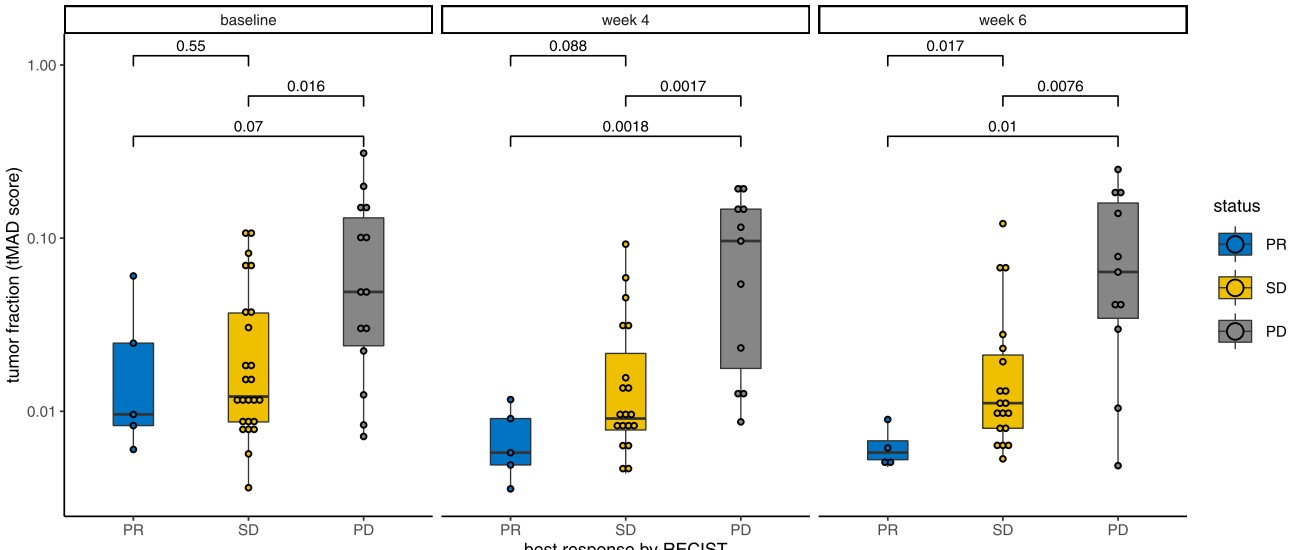

**Fig. 3 Modifications in the ctDNA tumour fraction measured with the t-MAD score from the baseline and early post-treatment samples (~week 4 and week 6).** p values for the Wilcoxon tests are indicated above the boxplots. Samples are separated based on the best overall response on RECIST (blue = PR, yellow = SD, grey = PD). There were n = 44 biologically different samples at baseline, 35 at week 4 and 34 at week 6. All samples are indicated by dots. Boxplot centre is the median, bounds of box represent 25 and 75 percentile, lines are 10 and 90 percentile. Source data are provided as a Source Data file.

In conclusion, we observed that FGFR pathway inhibition in a group of unselected ER positive women with aromatase inhibitor-resistant metastatic breast cancer showed some signs of activity. However, there was significant toxicity and although this was mostly grade 1–2, the RPED and metabolic effects do need careful monitoring. We have identified a gene expression signature, which, if confirmed in other studies, could enable the benefit of endocrine therapy to be extended in patients whose cancers show this signature. Hopefully, in future trials in which patients enriched for a signature such as this are included will recruit larger patient numbers to improve the current modest response rate seen.

## Methods

A list of enrolling centres along with The Research and Development (R&D) Department at each participating centre is provided in Supplementary Table 1a. The protocol and all amendments for this study were reviewed and given favourable opinions by the NRES Committee East Midlands – Derby. Favourable opinions applied to all NHS sites participating in the study, subject to permission being obtained from each NHS site R&D office prior to them starting in the study and using new versions of the protocol.

This study was conducted in accordance with the ethical principles outlined in the Declaration of Helsinki. Patients were informed about their right to withdraw from the trial. A written and signed confirmation stating that the patient had been adequately informed about the study and accepted to participate was obtained from each patient prior to the performance of any study specific procedure and registration into the study.

**Patients.** Eligible patients were those with metastatic breast cancer whose cancers had progressed on treatment with anastrozole or letrozole, either in the adjuvant or first line metastatic (safety run-in only) or any setting (phase IIa only). The NSAI did not have to be the most recent line of treatment. Inclusion criteria included: written informed consent, ECOG performance status 0–1; ≥25 years of age, postmenopausal with histologically-confirmed ER+ breast cancer (primary or metastatic tumour tissue) with at least 1 lesion that could be accurately assessed by CT/MRI/x-ray at baseline.

Exclusion criteria included treatment with any of the following: for the safety-run-in only, >1 regimen of endocrine therapy and >1 prior regimen of chemotherapy for advanced breast cancer and across the whole trial, previous exposure to any FGFR inhibitor, major surgery or radiotherapy within 4 weeks prior to first dose of study treatment and taking potent inhibitors or inducers of CYP3A4 or CYP2D6. Any of the following cardiac criteria precluded patients entering: Mean resting QTc interval >470 msec; any clinically important abnormalities in rhythm, conduction or morphology; factors that increase the risk

of QTc prolongation or risk of arrhythmic events; inadequate bone marrow liver or renal function or elevated calcium or phosphate or significant gastrointestinal disorders. Any of the following ophthalmological criteria were also exclusion criteria: Current evidence or previous history of retinal pigmented epithelium detachment (RPED); Previous laser treatment or intra-ocular injection for treatment of macular degeneration; Current evidence or previous history of soft drusen, drusenoid RPE detachment and wet macular degeneration; Current evidence or previous history of retinal vein occlusion (RVO); Current evidence or previous history of retinal degenerative diseases (e.g. hereditary). Patients with uncontrolled glaucoma or intra-ocular pressure >21 mmHg at screening were referred for ophthalmological management and the condition controlled prior to first dose of study treatment.

**Study design.** This was a phase IIa (with safety run-in), multi-centre, open label, single arm study of AZD4547 administered orally together with anastrozole or letrozole in ER + MBC breast cancer patients. The primary objective of the safety run-in component was to assess the safety and tolerability of AZD4547 (80 mg twice daily as a starting dose-Cohort 1; and we intended if required to follow this by <80 mg twice daily-Cohort 2) in combination with anastrozole (1 mg) or letrozole (2.5 mg) daily respectively. A cohort was expected to have a minimum of 3 and maximum of 6 patients. The Safety Review Committee (SRC) were tasked to determine the dose de-escalation scheme and whether further cohorts were required to select the safe and tolerated dose of AZD4547 to be used in the phase IIa study. A total of 6 patients were recruited between 23 October 2012 and 07 August 2013.

In the Phase IIa study, we intended to enrol 50 patients to our combination treatment. These individuals had cancers who had progressed on their NSAI and either directly continued on this NSAI or re-started the same NSAI they had previously failed after a number of interval other systemic therapies. Subjects received 80 mg of AZD4547 twice daily, one week on and one week off in addition to the daily NSAI of either letrozole (2.5 mg) or anastrozole (1 mg) which their cancers had progressed on. Treatment continued until either further disease progression or the development of unacceptable toxicities. Patient recruitment is provided in the CONSORT diagram (Supplementary Fig. 1). Patient recruitment took place between 24 April 2014 and 11 December 2015.

The choice of the AZD4547 schedule was selected as it had been well tolerated in 2 prior studies[6,8]. However, if 2 or more cases of severe toxicity (leading to permanent discontinuation of study drug) were observed in the first 6 patients, the Independent Data Monitoring Committee (IDMC) were to consider an alternative schedule, depending on emerging data from other AZD4547 studies. Secondary Objectives of the Safety run-in were to assess the pharmacokinetics (PK) of anastrozole or letrozole when given alone compared to in combination with AZD4547 and to describe the PK of AZD4547 when given in combination with anastrozole or letrozole.

The primary objective of the phase IIa study was to assess the activity of AZD4547 based on the change in tumour size at 12 weeks using RECIST criteria (or progression if prior to week 12), when used in combination with either

anastrozole or letrozole in ER positive breast cancer patients who have progressed on treatment with either anastrozole or letrozole in any setting. The 12-week timepoint was decided upon after consideration of the need to assess adverse events in this study population in relation to potential benefit. Patients would continue the AI that had previously been administered. Secondary objective of the Phase IIa study were to assess the activity of AZD4547 in combination with anastrozole or letrozole as measured by tumour response (RECIST criteria)[18]: at 6 weeks, 12 weeks, then every 8 weeks, and effect on progression-free survival (PFS). In addition, we aimed to assess the safety and tolerability of AZD4547 in combination with anastrozole or letrozole.

AZD4547 was supplied free of charge by AstraZeneca for use in the clinical trial as an Investigational Medicinal Product (IMP) from seven batches: P/5406/25; P/5406/47; 11-002792AZ; 14-002508AZ; 14-000425AZ; 15-001026AZ and L006495.

Details of Trial management are shown in Supplementary Table 1c. The study was registered with the European Clinical Trial Register on 5/1/2012 (2011-000454-32) and on the ISRCTN Register (80307982) on 27/04/2012. The study was also registered on ClinicalTrials.gov (NCT01791985) and was initially published on 13/02/2013. The protocol is available at http://www.imperialclinicaltrialsunit.org/trials/ and as Supplementary Note 2 in the Supplementary Information file.

**Safety and efficacy assessments**. Adverse events were recorded as per the NCI Common Toxicity Criteria for Adverse Events (version 4.3). The patient monitoring plan is detailed in Supplementary Table 1b. An ophthalmic assessment was performed by an ophthalmic expert at screening, Cycle 2 day 1, Cycle 3 day 1 and Cycle 4 day 1 (+/− 3 days). Thereafter, patients continuing the study treatment received a full ophthalmological review every 8 weeks (+/− 1 week) and finally at the study discontinuation visit. At any other time, abnormal visual symptoms or signs triggered a full ophthalmological review. Management of patients took into account the corrected calcium result and corrected according to the calcium:-phosphate ratio. If a patient experienced a doubling of phosphate levels from baseline or a calcium:phosphate product >4.5 mmol/L then phosphate chelation therapy was initiated, and clinical chemistry monitored weekly until resolution of the parameter to below the intervention limit (Fig. 1). Cardiac monitoring using Electrocardiogram and Echocardiography occurred at baseline, and 3 monthly thereafter until the end of the study.

Central review of subject's CT and MRI scans by RECIST was undertaken by an independent radiologist: Dr Adrian Lim, Imperial College Healthcare NHS Trust. Patients were permitted to stay on study treatment until progression of disease or severe toxicity. Details are in Supplementary Tables 2 and 3).

**Pharmaco-kinetic (PK) analyses**. Samples were obtained from patients in the safety run-in component. Venous blood samples (2 ×2.7 mL) for determination of concentrations of AZD4547, anastrozole and letrozole in plasma was taken at the times presented in Supplementary Table 1b on both day 7 of the NSAI monotherapy and day 7 of cycle 1 of the AZD4547 + anastrozole or letrozole combination therapy. These plasma samples were also analysed for phosphate and the data used with the AZD4547 PK data to investigate the PK / PD relationship. Samples for determination of AZD4547 concentrations in plasma were analysed at PRA International, The Netherlands. Samples for determination of anastrozole and letrozole concentrations in plasma were analysed at Covance, UK, using appropriate bioanalytical methods. Details are shown in the accompanying Covance document, available as Supplementary Note 1 in the Supplementary Information file.

**Trial statistics and analysis**. Justification of sample size: With a power of 85% and type I error of one-sided 0.05, 4 or more patients with clinical benefit out of 20 patients was the calculated criteria for the study to safely continue. The required total sample size for phase IIa assuming a standard deviation of 0.30 for the change in tumour size was 50 patients. The 95% confidence interval for the observed geometric mean will extend 1.09 in either direction with the total sample size. For the primary analysis, a 95% two-sided confidence interval were calculated for change in tumour. The study data were summarised using standard descriptive methods. Histograms and box-plots were used to assess the distributional assumptions and to check for possible outliers. Continuous variables that follow an approximately normal distribution were summarised using the mean and SD. Skewed continuous variables were summarised using the median and inter-quartile range (IQR). Categorical variables (binary, ordered and multinomial) are presented in terms of frequencies and percentages. The analysis of change in tumour size was performed on complete cases only, i.e. patients with tumour size measured at each time point until week 28. The secondary analyses of tumour response, progression free survival (PFS) and safety analysis were done on all patients who receive at least one dose of study treatment. No imputation for missing data was undertaken. Data were recorded using the InForm (version 4.6) electronic data capture (EDC) and management system and were analysed using Stata 16.1 (StataCorp. 2019. Stata Statistical Software: Release 16. College Station, TX: StataCorp LLC) and R software version 4.0 (R 4.0.0 (R Core Team (2020). R: A language and environment for statistical computing. R Foundation for Statistical Computing, Vienna, Austria. URL https://www.R-project.org/).

**Sample collection**. Blood samples were collected in the different clinical samples, and plasma isolated within 2 h following previously established procedure. Briefly, DNA was extracted from 2 to 4 mL of plasma using the QIAamp circulating nucleic acid kit (Qiagen) or QIAsymphony (Qiagen) according to the manufacturer's instructions. Samples were eluted in respectively 50 μL or 70 μL of elution buffer. Internal oligonucleotide controls, based on the Xenopus Tropicalis genome, were spiked in the samples to estimate the efficiency of DNA extraction (Forward PCR primer - 5′-GTGATCATGGGATTTGTAGCTGTT - 3′; Reverse PCR primer – 5′ AAACCAACCTGAAAACCATGGA - 3′).

**Shallow WGS**. Indexed sequencing libraries were prepared using a commercial kit (ThruPLEX-Plasma Seq, Takara). Pooled libraries (equimolar amounts) were sequenced with <1× depth of coverage using a HiSeq 4000 (Illumina), generating 150-bp paired-end reads. Paired end sequence reads were aligned to the human reference genome (GRCh37) using BWA-mem following removal of contaminating adapter sequences. PCR and optical duplicates were marked using MarkDuplicates (Picard Tools), and were excluded from downstream analysis, similarly to low mapping quality and supplementary alignments. When necessary, reads were down-sampled to 10 million in all samples for comparison purposes.

**Somatic copy number aberration analysis**. Analysis was performed in R using CNAclinic (https://github.com/sdchandra/CNAclinic)[8]. Shallow WGS reads were randomly sampled to 10 million reads per dataset and allocated into 30 Kbp non-overlapping bins throughout the length of the genome. In each bin read counts were corrected for sequence GC content and mappability, and bins overlapping 'blacklisted' regions (as per ENCODE project + 1000 Genomes database) were excluded from downstream analysis. Read counts in test samples were normalised by those from an identically processed healthy individual and log2 transformed to obtained copy number ratio values per genomic bin. Read counts in healthy controls were median normalised. Bins were then segmented using both Circular Binary Segmentation and Hidden Markov Model algorithms, and an averaged log2R value per bin was calculated. An in-house empirical blacklist of aberrant read count regions was constructed as follows: 65 sWGS datasets from healthy plasma were used to calculate median read counts per 30 Kbp genomic bin as a function of GC content and mappability[8]. A 2D LOESS surface was applied, and the median of differences between actual counts and the LOESS fitted values was calculated. across the 65 controls for each genomic bin. Regions showing median values greater than 4 standard deviations were blacklisted. The averaged segmental log2R values in each test sample overlapping this cfDNA blacklist were trimmed, and the median absolute value was calculated. This score was defined as t-MAD or the trimmed median absolute deviation from log2R = 0. The R code to reproduce this analysis is provided in https://github.com/sdchandra/tMAD[8,19].

**HTG EdgeSeq of FFPE samples**. The HTG EdgeSeq Oncology Biomarker Panel was used to measure mRNA expression levels of genes associated with tumour biology. This was done using micro-dissected FFPE sections obtained at original disease presentation from 12 patients with stable (SD) and 6 with progressive disease (PD) in response to combined inhibition of aromatase and FGFRs. The HTG EdgeSeq system combines HTG's proprietary quantitative nuclease protection assay (qNPA) chemistry with a next-generation sequencing (NGS)-based platform to enable the semi-quantitative analysis of a panel of targeted genes in a single assay. Functional DNA nuclease protection probes (NPPs) are hybridised to target mRNAs. S1 nuclease is added to digest excess non-hybridised DNA probes and non-hybridised mRNA, leaving only NPPs hybridised to mRNA fully intact and able to be amplified and barcoded. This produces essentially a 1:1 ratio of DNA detection probes to the mRNA targets present in the sample. Lastly, the NPPs are quantified by NGS. The HTG EdgeSeq Oncology Biomarker Panel contains 2,568 NPPs that measure gene expression counts of target genes, specifically 2549 genes, ten External RNA Controls Consortium (ERCC) probes, four internal positive controls (POS), and four internal negative controls (ANTs). A fully annotated gene list for this assay can be accessed on the support section of our website at https://www.htgmolecular.com/support.

**Gene expression analysis and associated bioinformatics**. The gene expression data was median normalised and analysed in the HTG EdgeSeq Reveal software to highlight differentially expressed genes (DEGs) between patients with stable (SD) and those with progressive disease (PD) using either the DESeq2 or the edgeR algorithm and the two lists combined. The *ggplot2* package in R was used to plot the expression of individual genes in PD vs SD patients. The HTG EdgeSeq Reveal software was also used to perform hierarchical clustering and principal component analysis.

DEGs were imported into Cytoscape using the Reactome FI plugin for functional network building with continuous nodes colour mapping based associated fold changes in expression and nodes size mapping based on network betweenness centrality (calculated based on network analysis tools in Cytoscape). Gene ontology and pathway enrichment analysis for DEGs were performed in EnrichR.

Expression correlation analysis was performed in R with Spearman correlation and associated t-test calculated using the base cor.test function and heatmaps

plotted using heatmap.2 from the *gplots* package. Linear correlation analysis was performed using lmRob from the *robust* package. The GDC TCGA Breast cancer dataset (TCGA-BRCA.htseq_fpkm-uq) was downloaded from the Xena web interface while the METABRIC dataset (data_expression_median) was downloaded from cBioportal.

**Reporting summary**. Further information on research design is available in the Nature Research Reporting Summary linked to this article.

## Data availability

The full study protocol is available as Supplementary Note 2 in the Supplementary Information file. The gene expression data generated in this study have been deposited in the Gene Expression Omnibus (GEO) database under accession code GSE198650. The clinical datasets generated during and/or analysed in the study which are not made publicly available due to data privacy laws can be made available upon request. Any requests for additional clinical data will be reviewed by the Imperial Clinical Trials Unit (Study Chief Investigator, Study Operations Manager, Head of Statistics, Q.A. Manager and Director of Operations). Any data to be shared will need a Data Sharing Agreement in place. All data shared will be de-identified. Data will be sharable as soon as possible after any Data Sharing Agreements are in place. The timeline will depend on the amount and format of the data requested. The data will be available for at least 10 years. All clinical data request should be addressed to Raoul Charles Coombes (c.coombes@imperial.ac.uk) and Michael Seckl (m.seckl@imperial.ac.uk). The remaining data are available within the Article, Supplementary Information file and Source data file.

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

## Acknowledgements

We thank all the RADICAL trial participants and the women who participated in this study, the research teams at participating hospitals (Russells Hall Hospital and Queen's Hospital Burton), the ECMCs (Imperial College London, Glasgow, Cambridge, Newcastle and Manchester) and the ECMC Combinations Alliance for their support. We also acknowledge the NIHR Manchester Clinical Research Facility at The Christie NHS Foundation Trust. This work was supported by Cancer Research UK (C1312/A12956) and AstraZeneca (D9010C00011) as part of the Experimental Cancer Medicines Centre (ECMC), DoH and CRUK Combinations Alliance initiative. IMP (AZD4547) was provided free of charge by AZ and distributed by Fisher Clinical Services. The study was Sponsored by Imperial College London and coordinated by the Imperial Clinical Trials Unit – Cancer, Imperial College London (ICTU-Ca) on behalf of the Sponsor. ICTU-Ca were involved in study design, data collection, analysis and manuscript writing. The authors thank ICTU Clinical Data Systems team for designing the eCRFs and database capture system. ICTU–Ca receives infrastructure support from Imperial Experimental Cancer Medicine Centre, Cancer Research UK Imperial Centre, National Institute for Health Research (NIHR) Imperial Biomedical Research Centre (BRC) and Imperial College Healthcare NHS Trust Tissue Bank. The study was also supported by the CRUK Cambridge Centre, the Cambridge ECMC and NIHR BRC and Cambridge Clinical Research Centre. We also thank the independent members of the trial steering committee (Prof Robert Coleman (Chair); Prof. Chris Twelves and Dr Janine Mansi) and the independent data monitoring committee (Prof. Daniel Rea (Chair); Dr Shah-Jalal Sarker and Dr Larry Hayward). We are also extremely grateful to the Independent Cancer Patients' Voice, particularly Adrienne Morgan for her input, guidance and help with the patient information sheet and to April Ann Matthews for her participation in the study steering committee meetings. The views expressed are those of the author(s) and not necessarily those of the NHS, the NIHR or the Department of Health.

## Author contributions

All authors have contributed to drafting and review of the manuscript. R.C.C., M.J.S., R.D.B., X.L., D.L., I.R.M., L.M., R.P and V.F. were responsible for the trial design. Molecular biomarker work and its analysis were performed by O.E.P., N.R., F.M., J.G.C. and R.D.B. Clinical and biomarker statistical analyses were performed by J.P.L.K., X.L., F.M. and OEP. Patient recruitment, management and data collection were provided by R.C.C., I.R.M., I.Z., R.D.B., N.C., R.P., A.A., R.A. Central imaging review was performed by A.L. and trial management was provided by P.D.B. with oversight from H.N.

## Competing interests

N.R. is co-founder and officer of Inivata, a company that commercialises technologies for ctDNA analysis. D.L. consults to Astrazeneca (AZ) and was an employee in the past. His research team has also received grant funding from AZ. He also has a Director role and is an employee of his own consultancy company DeLondra Oncology Ltd. RCC currently has a grant from AZ. I.M. consults to AZ. R.B. consults to AZ and has received grant funding, travel, accommodation, expenses from AZ. A.A. has received grant funding to her Institution from AZ. P.B., I.Z., X.L., J.P.L.K., H.N., M.S., N.C., E.P., L.M., A.L., F.M., O.P., R.A., J.G.C. and V.F. have indicated no competing interests.
