## [Peer Review File · Nature Communications]

Reviewers' Comments:

Reviewer #1:

Remarks to the Author:

The paper reports the safety and efficacy results of the RADICAL trial, as well as the exploratory biomarker analysis. Detail is given for Safety run-in cohort (including stopping rule) and for adverse event. Below is a list of concerns of efficacy and biomarker analysis for clarification.

1. Discrepant description of primary endpoint between manuscript and protocol: The protocol for phase IIA lists 'Change in tumour size at 12 weeks (or progression if prior to week 12)' as the primary endpoint and 'Tumour response RECIST criteria with 4 categories: complete response (CR), partial response (PR), stable disease (SD), progressive disease (PD)' as a secondary endpoint. However, the manuscript combines both as the primary endpoint: 'The primary objective of the phase Ila study was to assess the activity of AZD4547 based on the change in tumour size at 12 weeks using RECIST criteria (or progression if prior to week12)'. Clarification is needed because both definitions could lead to different conclusion. For example, a patient with SD could mean either 10% increases or 10% reduction. Based on RECIST, both 10% increase or reduction are considered as SD. However, if based on the tumor size metric, it could lead to opposite direction depending on its increase or reduction.
2. Confusion of sample size justification: In one hand, interim analysis is used for justification of 4 out of 20 with clinical benefit for trial continuation, but lack of statistical hypothesis to support the 85% power and 5% type I error. On the other hand, standard deviation of change of tumor size is used to justify the sample size of 50 patients. However, the result section did not report any 95% confidence interval for the change of tumor size. Instead, 95%CI for response rate was given.
3. Lack of interim analysis report: the trial includes an interim analysis, but no report in the manuscript.
4. The report of change of tumor size at 12 weeks (Table S4) should be considered as a primary table, instead of a supplementary table, because the 12 weeks is the time cutoff for the primary endpoint. Also, a 'Swimmer Plot' for the 12 weeks should be included as a primary plot in addition to the one for ORR. Similarly, Clinical Efficacy section should cover the result for change of tumor size at 12 weeks.
5. Lack of literature review to discuss whether the 26% Clinical Benefit Rate, 10% response rate, or 134 days of median PFS is clinically relevant.
6. The overall change of tumor size (proportion of tumor size change from baseline as the primary endpoint) is 0.18, indicating increase of tumor size. What is the implication of the result?
7. Significant biomarker analyses in various platforms were reported (e.g., ctDNA and serum phosphate) but without clear information of sample size. Also, it is unclear the conclusion of the analysis results.

Reviewer #2:

Remarks to the Author:

Overall, very well written manuscript on an important new agent. This reviewer appreciates the balanced presentation of the data and the thoughtful discussion of the largely negative results from the biomarker discovery analysis. A large part of this failure may be due to the fact that the samples analyzed were not taken at the time the patients were treated (i.e. were primaries, likely accrued years before metastatic disease arose) (16 primary and 4 metastatic samples taken at diagnosis). I think the authors might like to make a statement regarding the future need for a metastatic disease biopsy to be made available when ever feasible in studies such as this to progress the biomarker research more effectively.

Reviewer #3:

Remarks to the Author:

The manuscript by Coombes et al. on "A single arm phase IIa study (with combination safety 2 run-in) to assess the safety and efficacy of AZD4547 in combination with 3 either anastrozole or letrozole in patients with ER positive breast cancer 4 who have progressed on treatment with

anastrozole or letrozole – 5 RADICAL”

Investigated the FGFR1 inhibitor AZD4547 in a phase IIa study with a safety run in.

Overall it is an interesting and results rich paper but unfortunately with limited results to further work with.

Abstract:

Results section does not state the baseline.

Conclusion in the abstract section does not mirror the one in the main body of the text. Prefer the more modest version in the main body. 3/50 responses is not a great achievement and given the toxicity I would not support further developing this drug.

Overall the abstract should reflect the main results and conclusion.

Main body

Some of the methods could be in the supplement

Results section is somehow unclear. Please restructure it more clearly.

Would prefer to have some statement of baseline characteristics and reference to the table in the main text. From the baseline table it is not clear whether the prior treatments were conducted at the time of primary diagnosis and are part of treatment for early breast cancer. The rate of missing dates is very high. In about 25% of the patient stage, ER status in a third of the patients is missing. Please elaborate.

How were these missing patients classified in the further analysis. Could this be the result of the outcome.

Discussion should clearly state the primary endpoint of response rate at weeks 12. The discussion is of more general nature and could discuss the results a bit more.

Suggest to add in the discussion a section comparing the different FGFR drugs. Some clearly have other key toxicities but none seems really needed in breast cancer.

Please add a section of limitations

Minor:

Use generic names

Page 8 line 289 (Eisenhauer et al. Can be deleted). There are other parts with small mistakes.

Overall it is a bit unkempt.

Reviewer #4:

Remarks to the Author:

Review of Coombes, et al.

In this manuscript, the authors present the results and correlative analyses of the RADICAL study, a single arm phase IIa study of FGFR inhibitor AZD4547 with aromatase inhibitor in metastatic ER+/HER2- breast cancer.

The strengths of this manuscript include targeting of a frequently altered pathway in ER+/HER2- breast cancer (also known to be a mechanism of resistance to AI+CDK4/6 inhibitors), clear presentation of trial results, and interesting correlative analyses.

The limitations of this study include a relatively modest disease control and/or response rate and hypothesis generating correlative analyses that are not yet developed enough to guide future therapy.

In terms of potential clinical impact, there may be activity to this agent but clinical utility would require further biomarker development to optimally identify a population, further characterization of the retinal detachment issues, and consideration of the complex treatment landscape of advanced ER+/HER2- breast cancer.

Overall, while interesting, this manuscript likely is more appropriate for an oncology and/or breast cancer specific journal due to the lack of deep mechanistic or very clear biomarker identification.

MAJOR COMMENTS:

-Clinical trial results, efficacy, and adverse events clearly reported.

-ctDNA sWGS: The authors suggest that high tumor fraction via t-MAD could be used in the future to switch to alternate therapies. In Fig 3, at baseline the SD/PR patients have low tumor fraction. This could suggest that the tumor fraction is a biomarker of more indolent disease, not necessarily effectiveness of this agent or combination.

-6-gene signature: The authors use the HTG EdgeSeq technology to evaluate expression of ~2500 genes. Through multiple steps to optimize the fit, they land on 6 genes. I would recommend moderating the comments about these DEGs as the authors clearly state that they effectively fit the data to the outcome (multiple PCA approaches, etc). This raises significant questions about the likelihood of reproducibility, particularly in the absence of a validation cohort. More appropriately, this is a hypothesis generating approach and these 6 genes demonstrate a starting point for further biomarker development.

-Context of FGFR inhibition in metastatic breast cancer: The authors spend most of the discussion rehashing results. It would be more valuable to develop the context of FGFR inhibitors. The authors provide brief discussion of prior studies that show limited efficacy. However, they fail to mention the growing literature suggesting that (among many possible resistance mechanisms), FGFR alterations and pathway activation is likely a frequent mechanism of resistance to AI+CDK4/6i (Croessmann et al., 2019, Formisano et al., 2019, Mao et al., 2019, Nayar et al., 2019, Wander et al., 2019)

MINOR COMMENTS:

-Fig 1 (phosphate change) would be better presented with each week as a segment of the x-axis rather than color, which is difficult to track (and for red-green color blind individuals).

-Retinal detachment. Appreciate the table regarding retinal detachment. This trial did require multiple ophthalmologic examinations and, while the authors downplay the significance of PRED in the discussion as mostly asymptomatic, the frequency of examination could prove complex for further development of this agent.

REVIEWER COMMENTS

Reviewer #1 (Remarks to the Author):

The paper reports the safety and efficacy results of the RADICAL trial, as well as the exploratory biomarker analysis. Detail is given for Safety run-in cohort (including stopping rule) and for adverse event. Below is a list of concerns of efficacy and biomarker analysis for clarification.

1. Discrepant description of primary endpoint between manuscript and protocol: The protocol for phase IIA lists 'Change in tumour size at 12 weeks (or progression if prior to week 12)' as the primary endpoint and 'Tumour response RECIST criteria with 4 categories: complete response (CR), partial response (PR), stable disease (SD), progressive disease (PD)' as a secondary endpoint. However, the manuscript combines both as the primary endpoint: 'The primary objective of the phase IIA study was to assess the activity of AZD4547 based on the change in tumour size at 12 weeks using RECIST criteria (or progression if prior to week12)'. Clarification is needed because both definitions could lead to different conclusion. For example, a patient with SD could mean either 10% increases or 10% reduction. Based on RECIST, both 10% increase or reduction are considered as SD. However, if based on the tumor size metric, it could lead to opposite direction depending on its increase or reduction.

Response:

We kept the following throughout the study:

Primary Endpoint: Phase IIA:

Change in tumour size at 12 weeks (or progression if prior to week 12). The 12-week time point was chosen because of safety concerns at the start of the study; the team wanted to ensure benefit outweighed adverse events at this time point.

Secondary Endpoints: Phase IIA:

Change in tumour size at 6 weeks, 20 weeks, then every 8 weeks, as per study plan.

The IDMC had wondered how reliable the 12 week RECIST data was for those patients with slow progression, and asked if they could review the 20 week RECIST data for patients remaining on the study. IDMC were happy with 20 week data and recommended that the study continued.

We have included the 12 week data in a revised Waterfall plot (Fig 2b) so the reader could see the outcome in each patient according to the primary and secondary endpoints.

The relevant section of the IDMC reports is reprinted here:.

17/4/2015: Closed report to the IDMC:

The interim statistical analysis (futility analysis) will focus on the efficacy of AZD4547 when used in combination with either anastrozole or letrozole and allow the Independent Data Monitoring Committee (IDMC) an opportunity to evaluate the 12-week clinical benefit and determine if the study should continue. The analysis will also review the safety data with the objective to identify any safety issues experienced with the treatment.

The interim analysis will be undertaken after the 20th patient in the RADICAL study has completed 12 week follow up.

An interim analysis will occur after 20 patients are recruited into the study (including 6 patients in the safety run in phase) and have completed their 12-week follow up. Recruitment into the study will continue whilst the analysis is being carried out.

$\geq 30\%$ will be considered as the 12-week clinical benefit rate (stable disease, partial response, complete response) of interest and $\leq 5\%$ as non-significant 12-week clinical benefit rate. With power of 85% and type I error of one-sided 0.05, the study will continue if 4 or more patients out of 20 show clinical benefit. If less than 4 patients show clinical benefit the study will stop.

In summary, in the first 20 patients of RADICAL, 6 patients showed 12-week clinical benefit and thus the 12-week clinical benefit rate is 30.0% (95%CI: 11.9% - 54.3%).

Among the 14 patients in the PIIa, there were 180 AEs and no SAE was observed.

The decision was made that the study should therefore continue.

2. Confusion of sample size justification: In one hand, interim analysis is used for justification of 4 out of 20 with clinical benefit for trial continuation, but lack of statistical hypothesis to support the 85% power and 5% type I error. On the other hand, standard deviation of change of tumor size is used to justify the sample size of 50 patients. However, the result section did not report any 95% confidence interval for the change of tumor size. Instead, 95%CI for response rate was given.

Response:

The justification on the sample size is specified in the manuscript:

"With a power of 85% and type I error of one-sided 0.05, 4 or more patients with clinical benefit out of 20 patients was the calculated criteria for the study to safely continue. The required total sample size for phase IIa assuming a standard deviation of 0.30 for the change in tumour size was 50 patients. The 95% confidence interval for the observed geometric mean will extend 1.09 in either direction with the total sample size. For the primary analysis, a 95% two-sided confidence interval were calculated for change in tumour."

Table S4 (now Table 3 in the revised Tables) has been updated to include the 95%CI as well as the range for the change in tumour size.

3. Lack of interim analysis report: the trial includes an interim analysis, but no report in the manuscript.

Response:

The relevant points of the interim analyses are included here:

1.2.1. Interim Analysis

An interim analysis will occur after 20 patients are recruited into the study (including patients in the safety run in phase) and have completed their 12-week follow up. Following recommendations from the Independent Data Monitoring Committee (IDMC) data will also be reviewed once 20 patients

(including patients in the safety run in phase) have completed their 20-week follow up. Recruitment into the study will continue whilst the analysis is being carried out.

≥30% will be considered as the 12/20-week clinical benefit rate (stable disease, partial response, complete response) of interest and ≤5% as non-significant 12/20-week clinical benefit rate. With power of 85% and type I error of one-sided 0.05, the study will continue if 4 or more patients out of 20 show clinical benefit. If less than 4 patients show clinical benefit the study will only continue if a particular biomarker has been identified.

4. The report of change of tumor size at 12 weeks (Table S4) should be considered as a primary table, instead of a supplementary table, because the 12 weeks is the time cut-off for the primary endpoint. Also, a 'Swimmer Plot' for the 12 weeks should be included as a primary plot in addition to the one for ORR. Similarly, Clinical Efficacy section should cover the result for change of tumor size at 12 weeks.

Response:

We have added the table to the main tables (Table 3) and added the week 12 data to the swimmer plot. (Fig 2b).

5. Lack of literature review to discuss whether the 26% Clinical Benefit Rate, 10% response rate, or 134 days of median PFS is clinically relevant.

We have discussed this now more fully, and pointed out the limits of the treatment, but also respectfully suggest that, with methods that can be used to enrich for a population that could benefit from AZD4547, a higher response rate might be observed.

6. The overall change of tumor size (proportion of tumor size change from baseline as the primary endpoint) is 0.18, indicating increase of tumor size. What is the implication of the result?

Response:

The change in tumour size can range from -1 (-100% or complete response) to positive infinity. Proportion of tumour size change of 0 (or 0%) means no change in tumour size from the baseline measurement. As expected, the results follow a positively skewed distribution in all the follow-up weeks, i.e. increase in tumour size compared to baseline values. On the average there is a 0.18 proportion of tumour size change from baseline to week 12. However, some patients experienced decreased tumour size as shown by the negative values in the range.

As explained above, in view of the relatively early stage in the development of AZD4547, we wanted to know at an early time point when patients had overt disease progression or lack of tolerability to the treatment that meant they needed to change therapy hence the importance of the clinical benefit measure which is why we included the latter in the analysis.

7. Significant biomarker analyses in various platforms were reported (e.g., ctDNA and serum phosphate) but without clear information of sample size. Also, it is unclear the conclusion of the analysis results.

Response:

We had phosphate and ctDNA results in all patients: this statement has now been included to clarify this. With regard to the DEG analysis, the differently expressed genes considered showed significant difference between the two groups and that the top hits enabled clear separation between the

responders and non-responders in unsupervised analysis (PCA). Higher sample numbers may have increased the number of genes in the list and improved the adjusted p-value but would not likely change the genes (number or nature) involved in the segregation.

Reviewer #2 (Remarks to the Author):

Overall, very well written manuscript on an important new agent. This reviewer appreciates the balanced presentation of the data and the thoughtful discussion of the largely negative results from the biomarker discovery analysis. A large part of this failure may be due to the fact that the samples analyzed were not taken at the time the patients were treated (i.e. were primaries, likely accrued years before metastatic disease arose) (16 primary and 4 metastatic samples taken at diagnosis). I think the authors might like to make a statement regarding the future need for a metastatic disease biopsy to be made available when ever feasible in studies such as this to progress the biomarker research more effectively.

Response:

We have included this in the discussion now and agree that in future, metastatic biopsies should be taken at the start of treatment ; however in our experience many patients with difficult to access metastatic sites decline biopsies frequently.

Reviewer #3 (Remarks to the Author):

The manuscript by Coombes et al. on “A single arm phase IIa study (with combination safety 2 run-in) to assess the safety and efficacy of AZD4547 in combination with 3 either anastrozole or letrozole in patients with ER positive breast cancer 4 who have progressed on treatment with anastrozole or letrozole – 5 RADICAL”

Investigated the FGFR1 inhibitor AZD4547 in a phase IIa study with a safety run in.

Overall it is an interesting and results rich paper but unfortunately with limited results to further work with.

Abstract:

Results section does not state the baseline.

Conclusion in the abstract section does not mirror the one in the main body of the text. Prefer the more modest version in the main body. 3/50 responses is not a great achievement and given the toxicity I would not support further developing this drug.

Overall the abstract should reflect the main results and conclusion.

Response:

We have now added the 12-week result in the abstract also. However, as stated above, the 12-week Primary end-point was put in place to ensure patient safety of this novel combination; we have therefore also included the 28-week result which is in line with current practice.

Main body

Some of the methods could be in the supplement

Results section is somehow unclear. Please restructure it more clearly.

Would prefer to have some statement of baseline characteristics and reference to the table in the main text. From the baseline table it is not clear whether the prior treatments were conducted at the time of primary diagnosis and are part of treatment for early breast cancer. The rate of missing dates is very high. In about 25% of the patient stage, ER status in a third of the patients is missing. Please elaborate.

How were these missing patients classified in the further analysis. Could this be the result of the outcome.

Response:

We have restructured the paper to clarify the results of the study. We have now separated the treatment history of the primary from those who presented with metastatic disease in Table 1. ER status was available in 100% of patients as shown in the table.

Discussion should clearly state the primary endpoint of response rate at weeks 12. The discussion is of more general and could discuss the results a bit more.

Suggest to add in the discussion a section comparing the different FGFR drugs. Some clearly have other key toxicities but none seems really needed in breast cancer.

Please add a section of limitations

Minor:

Use generic names

Page 8 line 289 (Eisenhauer et al. Can be deleted). There are other parts with small mistakes. Overall it is a bit unkempt.

Response:

We have done this in the discussion, deleting the reference and adding limitations and adverse effects of the different inhibitors. We have tidied up the document.

Reviewer #4 (Remarks to the Author):

Review of Coombes, et al.

In this manuscript, the authors present the results and correlative analyses of the RADICAL study, a single arm phase IIa study of FGFR inhibitor AZD4547 with aromatase inhibitor in metastatic ER+/HER2- breast cancer.

The strengths of this manuscript include targeting of a frequently altered pathway in ER+/HER2- breast cancer (also known to be a mechanism of resistance to AI+CDK4/6 inhibitors), clear presentation of trial results, and interesting correlative analyses.

The limitations of this study include a relatively modest disease control and/or response rate and hypothesis generating correlative analyses that are not yet developed enough to guide future therapy.

In terms of potential clinical impact, there may be activity to this agent but clinical utility would require further biomarker development to optimally identify a population, further characterization of the retinal detachment issues, and consideration of the complex treatment landscape of advanced ER+/HER2- breast cancer.

Overall, while interesting, this manuscript likely is more appropriate for an oncology and/or breast cancer specific journal due to the lack of deep mechanistic or very clear biomarker identification.

MAJOR COMMENTS:

-Clinical trial results, efficacy, and adverse events clearly reported.

-ctDNA sWGS: The authors suggest that high tumor fraction via t-MAD could be used in the future to switch to alternate therapies. In Fig 3, at baseline the SD/PR patients have low tumor fraction. This could suggest that the tumor fraction is a biomarker of more indolent disease, not necessarily effectiveness of this agent or combination.

Response:

We have re-worded this to clarify: The ctDNA tumor fraction (via t-MAD or any other mutation marker) correlates with the tumor volume as this was previously demonstrated (cf Abbosh et al, Nature, 2017; Zviran et al, Nat Med) and can reflect treatment response, but the tumor fraction (TF) alone (on single timepoint) is not automatically reflecting treatment response, but an analysis of the TF on multiple timepoints can reflect effectiveness of the agent (via a decrease in TF as indicated by the significant p values at week 6 and 8 in our plot, in comparison to baseline). We have also added a word of caution in the discussion.

-6-gene signature: The authors use the HTG EdgeSeq technology to evaluate expression of ~2500 genes. Through multiple steps to optimize the fit, they land on 6 genes. I would recommend moderating the comments about these DEGs as the authors clearly state that they effectively fit the data to the outcome (multiple PCA approaches, etc). This raises significant questions about the likelihood of reproducibility, particularly in the absence of a validation cohort. More appropriately, this is a hypothesis generating approach and these 6 genes demonstrate a starting point for further biomarker development.

Response:

We have added the following in the discussion:

. . . However, the reproducibility of this signature needs to be confirmed in a validation cohort. Hence, it should currently be viewed as a starting point for further biomarker development and future studies should optimally be carried out using biopsies obtained before commencing the FGFR inhibitor. However, the fact that expression correlations between these genes and other DEGs are conserved between the TCGA or METABRIC datasets and our own cohort suggests that the expression of these genes is not particularly biased in our patient population, which should maximise chances of future validation.

-Context of FGFR inhibition in metastatic breast cancer: The authors spend most of the discussion rehashing results. It would be more valuable to develop the context of FGFR inhibitors. The authors provide brief discussion of prior studies that show limited efficacy. However, they fail to mention the growing literature suggesting that (among many possible resistance mechanisms), FGFR alterations and pathway activation is likely a frequent mechanism of resistance to AI+CDK4/6i (Croessmann et al., 2019, Formisano et al., 2019, Mao et al., 2019, Nayar et al., 2019, Wander et al., 2019)

Response:

We have now added a recent reference that covers these points and altered the discussion and included discussion concerning CDK4/6 inhibitors as well as 2 of the references suggested.

MINOR COMMENTS:

-Fig 1 (phosphate change) would be better presented with each week as a segment of the x-axis rather than color, which is difficult to track (and for red-green color blind individuals).

-Retinal detachment. Appreciate the table regarding retinal detachment. This trial did require multiple ophthalmologic examinations and, while the authors downplay the significance of RPED in the discussion as mostly asymptomatic, the frequency of examination could prove complex for further development of this agent.

Response:

We have redrafted Fig 1 in a single colour. We have added more discussion regarding the issue of RPED and the consequence in the text.

Reviewers' Comments:

Reviewer #1:

Remarks to the Author:

the statistical issues have been addressed.

Reviewer #4:

Remarks to the Author:

The authors have appropriately responded to my critiques.